# Metabolism of Acetaminophen by Enteric Epithelial Cells Mitigates Hepatocellular Toxicity In Vitro

**DOI:** 10.3390/jcm12123995

**Published:** 2023-06-12

**Authors:** Katie Morgan, Steven D. Morley, Arslan K. Raja, Martin Vandeputte, Kay Samuel, Martin Waterfall, Natalie Z. M. Homer, Peter C. Hayes, Jonathan A. Fallowfield, John N. Plevris

**Affiliations:** 1Hepatology Laboratory, The University of Edinburgh, 49 Little France Crescent, Edinburgh EH16 4SB, UK; kmorgan1783@gmail.com (K.M.); steve.morley@ed.ac.uk (S.D.M.); a.raja-5@sms.ed.ac.uk (A.K.R.); m.f.r.vandeputte@sms.ed.ac.uk (M.V.); p.hayes@ed.ac.uk (P.C.H.); jonathan.fallowfield@ed.ac.uk (J.A.F.); 2Scottish Blood Transfusion Service, Jack Copland Centre, 52 Research Avenue North, Edinburgh EH14 4BE, UK; k.samuel@ed.ac.uk; 3Flow Cytometry Facility, Ashworth Laboratories, Institute of Immunology & Infection Research, The University of Edinburgh, The Kings Buildings, Edinburgh EH9 3FL, UK; martin.waterfall@ed.ac.uk; 4Mass Spectrometry Facility, Centre for Cardiovascular Science, Queen’s Medical Research Institute, The University of Edinburgh, 47 Little France Crescent, Edinburgh EH16 4TJ, UK; n.z.m.homer@ed.ac.uk; 5Institute for Regeneration and Repair, Edinburgh BioQuarter, The University of Edinburgh, 4-5 Little France Drive, Edinburgh EH16 4UU, UK

**Keywords:** paracetamol, acetaminophen, gut–liver axis, tight junctions, Caco-2, HepaRG, enterohepatic circulation

## Abstract

The gut–liver axis is defined by dietary and environmental communication between the gut, microbiome and the liver with its redox and immune systems, the overactivation of which can lead to hepatic injury. We used media preconditioning to mimic some aspects of the enterohepatic circulation by treating the human Caco-2 intestinal epithelial cell line with 5, 10 and 20 mM paracetamol (N-acetyl-para-aminophenol; APAP) for 24 h, after which cell culture supernatants were transferred to differentiated human hepatic HepaRG cells for a further 24 h. Cell viability was assessed by mitochondrial function and ATP production, while membrane integrity was monitored by cellular-based impedance. Metabolism by Caco-2 cells was determined by liquid chromatography with tandem mass spectrometry. Caco-2 cell viability was not affected by APAP, while cell membrane integrity and tight junctions were maintained and became tighter with increasing APAP concentrations, suggesting a reduction in the permeability of the intestinal epithelium. During 24 h incubation, Caco-2 cells metabolised 64–68% of APAP, leaving 32–36% of intact starting compound to be transferred to HepaRG cells. When cultured with Caco-2-preconditioned medium, HepaRG cells also showed no loss of cell viability or membrane integrity, completely in contrast to direct treatment with APAP, which resulted in a rapid loss of cell viability and membrane integrity and, ultimately, cell death. Thus, the pre-metabolism of APAP could mitigate previously observed hepatotoxicity to hepatic tight junctions caused by direct exposure to APAP. These observations could have important implications for the direct exposure of hepatic parenchyma to APAP, administered via the intravenous route.

## 1. Introduction

The gut–liver axis is a dynamic communication system in which the intestinal epithelial cell barrier allows the transport of nutrients and other small molecule compounds, whilst forming the first line of defence against pathogens and undesirable toxins that might otherwise cross over into the portal vein [1,2,3]. There are two ways in which solutes and molecules can pass through the epithelial layer into the lamina propria. The first of these is via a ‘leaky pathway’, which allows large molecules to pass through the epithelial layer [4]. The second is a regulated small pore pathway, which is only opened by charged molecules and is modulated by tight junctions, along with the adherens and claudin protein families, which together make up the apical junction complex [5]. Tight junctions are defined as multi-protein junctional complexes, including actin cytoskeleton-binding proteins and adhesive transmembrane proteins that provide a seal between adjacent epithelial cells, forming a barrier through which selective transport may be permitted and are often referred to as the “rate-limiting step” of transepithelial passage [4,5,6]. As such, intestinal tight junction integrity is essential for the regulation of homeostasis within the gut and as a knock-on effect, the liver.

The small molecule analgesic paracetamol (*N*-acetyl-para-aminophenol; APAP), which can be administered orally or via intravenous infusion, is metabolised in the liver by CYP3A4, 1A2 and 2E1 pathways into glucuronide (47–62%) and sulphate (25–36%) derivatives [7]. A smaller portion (8–10%) is oxidised into N-acetyl-p-benzoquinone-imine (NAPQI), a toxic by-product of APAP metabolism that is normally inactivated by conjugation with glutathione (GSH) [7]. However, the depletion of hepatic glutathione reserves results in oxidative stress and NAPQI-mediated tissue damage that stimulates macrophage recruitment and release of proinflammatory mediators, mainly tumour necrosis factor alpha, which cause hepatotoxicity through the disruption of tight junctions and, ultimately, cell death [5,6]. Epithelial cells of the intestinal tract also display significant CYP1A2 and CYP2E1 activity [8], the products of which may reach the liver via the venous portal circulation [9,10]. It is therefore important to consider possible enhancement or attenuation of hepatotoxicity arising from enteric metabolism of APAP when administered via the oral route.

Previously, we determined the toxic effects of 5, 10 and 20 mM APAP on HepaRG in vitro cell cultures, characterised by their differentiation into an intrinsic co-culture of hepatocytes and cholangiocytes, the formation of tight junctions and CYP enzyme activity levels comparable to primary human hepatocytes [11,12,13] and showed that even subtoxic doses (5 mM) of APAP disrupted hepatic tight junctions [14]. It has also been shown that APAP can be absorbed by small intestinal epithelial cells via transcellular and paracellular pathways and alters intestinal microvillar and epithelial cellular topography, potentially disrupting intestinal epithelial cell tight junctions [15,16,17,18,19]. This raises the possibility of APAP causing a “leaky gut”, allowing the passage of microbes and external solutes into the portal vein, thereby increasing inflammation and causing damage to the liver [1,2].

The aim of this study was to ask three questions, namely, whether direct exposure of intestinal epithelial cells to APAP affects intestinal tight junctions, to what extent intestinal epithelial cells can metabolise APAP and if the products of intestinal epithelial cell metabolism are more or less toxic to the liver than direct exposure to APAP. To investigate the direct enteric effects of APAP, we used Caco-2 cells, an immortalised human colorectal adenocarcinoma cell line which spontaneously differentiates into a heterogeneous mixture of enteric epithelial cells. Crucially, Caco-2 cells exhibit both paracellular and transcellular transport, form monolayers, which exhibit polarity and tight junctions and express significant levels of CYP enzyme activity and glutathione, making them ideal as an in vitro model of the intestinal epithelial barrier [6,8,20,21,22].

In the current study, we replicated our previous experimental conditions used to treat HepaRG cells with APAP [14] by exposing Caco-2 epithelial cells to 5, 10 and 20 mM APAP for 24 h while assessing cell adhesion, cell–cell tight junction stability and membrane integrity in real time using electric cell-substrate impedance sensing (ECIS) [23,24]. Conditioned Caco-2 cell culture supernatants were then sampled to determine the remaining concentration of APAP and transferred directly onto differentiated HepaRG hepatic cells for a further 24 h in vitro cell culture and real-time monitoring. In contrast to the previously reported disruption of HepaRG tight junctions by low dose APAP [14], Caco-2 cell–cell tight junctions were clearly maintained at all APAP concentrations tested, with a trend towards tightening, which became significant at higher APAP concentrations, suggesting the possibility that the in vivo intestinal epithelial barrier is maintained and even strengthened in the presence of APAP. This led us to consider the potential adaptive roles of the gut–liver axis in mitigating APAP hepatotoxicity.

## 2. Materials and Methods

### 2.1. Cell Culture

Human Caco-2 (ECACC 86010202) cells obtained from the European Collection of Authenticated Cell Cultures, were cultured to confluence in minimum essential medium Eagle (MEME+; Sigma-Aldrich, Gillingham, Dorset, UK) with 20% fetal bovine serum (FBS, Life Technologies, Paisley, Renfrew, UK) on 8-well ibidi arrays with 10+ electrodes per well for electric cell-substrate impedance sensing (ECIS) measurements, and on Corning 96-well cell culture plates for other assays (5000 cells per cm^2^). Cells were monitored using ECIS until a plateau was observed (~day 10), denoting confluence and presence of tight junctions, before APAP was added. Cells were cultured between 8–10 days for cell viability assays. APAP was added in a mixture of 90% MEME+ with 20% FBS and 10% ADD620 shown to support viability of both Caco-2 and HepaRG cell types comparable to controls (cells in their own media).

Human HepaRG™ cells (HPR116; cryopreserved after differentiation) obtained from Biopredic International, Rennes, France, were cultured to confluence using suppliers’ protocols. In brief, Williams E Medium with GlutaMAX™ was used as the basal medium with supplements purchased from Biopredic International. Cells were seeded in General Purpose HepaRG^®^ medium (ADD670) on day 0 at 2.4 × 10^5^/cm^2^ into 8-well ibidi arrays with 10+ electrodes per well for ECIS measurements and on Corning 96-well cell culture plates for other assays. On day 3, medium was changed to HepaRG™ Maintenance and Metabolism Medium (ADD620) and renewed every other day. Cultures were monitored by ECIS as above for 8 days before exposure to conditioned media.

### 2.2. Drugs

A 4M solution of APAP (A7085, Sigma-Aldrich, Gillingham, Dorset, UK) was prepared in dimethyl sulfoxide and diluted to 5, 10 and 20 mM in mixed cell culture medium. Caco-2 cells were incubated with previously delineated concentrations of APAP (0, 5, 10 and 20 mM) for 24 h (Figure 1, Day 10). Concentrations of APAP were based on a 4 mM solution in vitro being comparable to 2 × 200 mg tablets in an average human adult. Caco-2-conditioned cell culture supernatant was then transferred onto fresh HepaRG cells for a further 24 h (Figure 1, Day 11). Incubation of APAP for 24 h on Caco-2 cells followed by 24 h incubation on HepaRG cells was chosen to enable comparison with direct addition of APAP to hepatic cells [14]. Furthermore, 24 h incubation could reflect certain clinical situations where, for example, there is staggered overdose with APAP. Cell viability and percentage APAP metabolised were also measured after the 24 h incubation periods.

### 2.3. PrestoBlue^®^ Live-Cell Viability and CellTiter-Glo^®^ ATP-Depletion Endpoint Hepatocellular Toxicity Assays

Cell viability was measured using PrestoBlue^®^ to assess mitochondrial function and Promega CellTiter-Glo^®^ to assess total ATP production. Neither viability assay in isolation provides a reliable indication of cell health, but when measured in parallel can be used to monitor metabolic activity and energy production. Following the 24 h treatment with APAP or conditioned media, 10% (*v*/*v*) PrestoBlue^®^ (A-13262; Life Technologies Paisley, Renfrew, UK) was added to cell culture medium and incubated for 30 min, before measuring the fluorescence signal on a GloMax Multi-Microplate Reader (Promega, Southampton, UK). Total cellular ATP levels were determined on the same cells using the CellTiter-Glo^®^ Luminescent Cell Viability Assay (G7570; Promega, Southampton, UK) and read on the GloMax multi-microplate reader.

### 2.4. Electric Cell-Substrate Impedance Sensing^®^ (ECIS^®^) Impedance-Based Assay

An ECIS^®^ instrument-based assay (Z-Theta 16-Well Array Station, Applied BioPhysics, Troy, NY, USA) was used to monitor real-time changes in tight junction integrity and basolateral adhesion and overall cell membrane capacitance, in response to APAP [23,24]. Impedance was monitored with measurements taken at 160 s intervals over a 500 Hz to 64 kHz frequency range, for up to 24 h of APAP treatment. Normalisation on 4 kHz low frequency and 64 kHz high frequency data points was performed using Excel Version 2303 (Microsoft, Redmond, WA, USA) and GraphPad Prism Version 8.3 (Graphpad Software, Boston, MA, USA), translating these into normalised, quantitative data on cell–cell tight junction integrity and cell-electrode adhesion (4 kHz resistance) and cell membrane capacitance (64 kHz capacitance) [23].

### 2.5. APAP and Metabolite Analysis by Liquid Chromatography Tandem Mass Spectrometry (LC-MS/MS)

MS grade methanol and water (Fisher Scientific, Loughborough, UK) and formic acid were from Sigma-Aldrich (Gillingham, UK). APAP was from Apollo (Denton, Manchester, UK). 3-cysteinyl-acetaminophen (APAP-CYS) and acetaminophen-glucuronide (APAP-GLU) were from CGeneTech (Indianapolis, IN, USA). Glutathione-acetaminophen (APAP-GSH), acetaminophen-sulphate (APAP-SUL), acetaminophen-d4 (APAP-d4) and acetaminophen-sulphate-d3 (APAP-SUL-d3) were from Santa Cruz Biotechnology Inc. (Heidelberg, Karlsruhe, Germany).

#### Sample Preparation and Analysis of APAP and Its Metabolites by LC-MS/MS

Targeted analysis of APAP and its metabolites in cell media samples was carried out following protein precipitation and followed by analysis by LC-MS/MS on an Acquity UPLC—QTrap 5500 system alongside calibration standards. Briefly, calibration standards (0.1–100 ng) and cell media samples were aliquoted into individual wells of a 96-well PPT+ plate (Biotage, Uppsala, Sweden), enriched with the internal standard solution (Acetaminophen-d4;APAP-d4; 10 ng and 10 ng APAP-SUL-d3). The Extrahera liquid handling robot (Biotage, Uppsala, Sweden) added acetonitrile (0.4 mL) to each well, after which, the solvent was passed through the extraction plate using positive pressure and the eluate reduced to dryness under nitrogen on an SPE-Dry 96 dual concentrator (Biotage, Uppsala, Sweden). The dried residue was resuspended in water/methanol (70:30; 100 µL), the plate sealed with a zone-free 96-well plate sealing film (Sigma-Aldrich, Gillingham, UK) and 10 µL was injected onto the LC-MS/MS.

All components were separated by liquid chromatography using an HSS T3 (150 × 2.1 mm; 1.8 µm; Waters, UK) column, protected by a Kinetex KrudKatcher^®^ (Phenomenex, Macclesfield, UK) held at 45 °C, using a mobile phase system consisting of 0.1% formic acid in water (A) and 0.1% formic acid in methanol (B) at a flow rate of 0.5 mL/min. Gradient elution was achieved with a total run time of 7.5 min from 5% to 95% B on an Acquity Classic UPLC (Waters, Wilmslow, UK) connected to a QTrap 5500 mass spectrometer (AB Sciex, Macclesfield, UK), operated in multiple reaction monitoring mode. The mass spectrometer was operated in polarity switching electro-spray mode (550 °C, 5.5 kV/−4.5 kV). The analytes APAP-Cys, APAP-Sul, APAP-Glu, APAP and APAP-GSH eluted at 4.0, 4.5, 4.6 and 4.7 min, respectively. In positive mode, MRM transitions were *m*/*z* 152 to 110.0, 92.9 at 23 and 31 V for APAP, *m*/*z* 156.0 to 114.0, 97.0 at 23 and 23 V for APAP-d4, *m*/*z* 271.1 to 140.0, 182.0 at 35 and 35 V for APAP-Cys and *m*/*z* 457.2 to 140 at 33 V for APAP-GSH. For *m*/*z* 229.8 to 107.0, 150.1 at 36 and 15 V for APAP-SUL, *m*/*z* 326.0 to113.0, 150.0 at 28V for APAP-Glu and *m*/*z* 233.0 to 109.5, 181.4 at 30 and 5 V for APAP-SUL-d3, respectively.

### 2.6. Statistical Analysis of Data

#### 2.6.1. Cell Viability Data Analysis

One-way ANOVA with post hoc Tukey test of multiple treatments was used to assess statistical significance of total ATP and PrestoBlue data. A *p*-value < 0.05 was taken to be statistically significant.

#### 2.6.2. Impedance Data Analysis

Impedance data were analysed using MATLAB^®^ Version R2022a (MathWorks, Natick, MA, USA). To analyse differences in impedance curves, we used normalised values, calculated as impedance values from the microelectrode confluent with Caco-2 or HepaRG cells, divided by its value at the challenge starting point (t = 0). A *p*-value < 0.05 was taken to be statistically significant.

#### 2.6.3. LC-MS/MS Data Analysis

The ratio of the peak area of the analyte and internal standard versus the amount of APAP and metabolites was plotted, and linear regression analysis was calculated from the peak area ratio using the linear regression analysis equation for each analyte generated in MultiQuant^®^ 3.2.3 (AB Sciex, Macclesfield, UK).

## 3. Results

### 3.1. Cell Viability

There was no statistically significant change in Caco-2 cell ATP production and live cells compared to control following 24 h treatment with 5–20 mM APAP at any APAP concentration tested, albeit there was a non-statistically significant trend towards reduced viability at the highest (20 mM) APAP concentrations (Figure 2, Step 1). This suggests that Caco-2 cell monolayers maintain their integrity for at least 24 h when treated directly with APAP. In contrast, in our previous studies, HepaRG cells treated directly with 5–20 mM APAP for 24 h showed a rapid loss of membrane integrity and cell viability and ultimately cell death [14]. However, when cell culture medium containing 5–20 mM APAP is ‘preconditioned’ by incubation for 24 h with Caco-2 cells and then transferred to HepaRG cells for a further 24 h, no such loss of cell viability or ATP production is seen (Figure 2, Step 2). This suggests that Caco-2 cells can metabolise or otherwise alter APAP to derivatives that are no longer toxic to hepatic cells.

### 3.2. Cellular Impedance Assay of Cell Adhesion, Cell–Cell Tight Junctions and Membrane Integrity

We reported previously that loss of HepaRG cell viability and ATP production following direct exposure to 5–20 mM APAP for 24 h was accompanied by an even earlier reduction in cell adhesion and loss of cell–cell tight junctions [14]. We therefore applied impedance-based assays to Caco-2 cells exposed to 5–20 mM APAP for 24 h and HepaRG cells treated for 24 h with Caco-2-preconditioned medium to determine if similar changes could be observed. This showed that Caco-2 cell adhesion, cell–cell tight junctions (Figure 3, Step 1, 4 kHz) and membrane integrity (Figure 3, Step 1, 64 kHz) were maintained for at least 24 h following direct treatment with APAP, confirming the maintenance of Caco-2 cell monolayer integrity in vitro and suggesting that the in vivo intestinal epithelial cell barrier might also be maintained in the presence of APAP. Intriguingly, Caco-2 cells exposed to APAP showed a trend towards increased tight junction integrity (4 kHz), which was statistically significant at higher (20 mM) APAP concentrations, (Figure 3, Endpoint fold change Step 1, 4 kHz). This suggests a ‘tightening’ of cell–cell interactions in the Caco-2 cell monolayer in vitro, raising the possibility that the in vivo intestinal epithelium might also be strengthened in response to APAP (Figure 3).

ECIS assay of HepaRG cells cultured for 24 h with Caco-2-preconditioned media previously containing 5–20 mM APAP showed no loss of cell adhesion or cell–cell tight junctions (Figure 4, Step 2, 4 kHz) or membrane integrity (Figure 4, Step 2, 64 kHz), albeit there was a minor disruption of tight junctions around 10 h after the addition of Caco-2-preconditioned media and a non-significant trend towards a concentration-dependent increase in membrane capacitance. This is consistent with the maintenance of membrane integrity and ATP production observed in cell viability assays, but in complete contrast to our previously published experiments where direct application of APAP to HepaRG cells caused a concentration-dependent loss of tight junctions [14]. Taken together, this supports the notion that Caco-2 cells have metabolised APAP to derivatives that are no longer directly or indirectly toxic to hepatic cells.

### 3.3. LC-MS/MS Analysis of APAP Metabolism during Caco-2 Cell Culture

To analyse the fate of APAP during Caco-2 cell culture, LC-MS/MS analysis of supernatants from APAP-treated Caco-2 cells was carried out to quantify APAP metabolites and the remaining amount of unmetabolised APAP subsequently transferred to HepaRG cells. Following 24 h incubation, more than 76% of APAP was metabolised by Caco-2 cells. Subtoxic (5 mM, 10 mM) APAP concentrations were very efficiently metabolised by the Caco-2 cells (>85%), while 20 mM APAP that is considered toxic to hepatic cells still showed a high rate of conversion (>76%) (Table 1). Metabolism of APAP via Caco-2 cells primarily used the sulfation and glucuronidation pathways, while a smaller percentage was also metabolised via the cysteine pathway (Table 2).

Supernatant was collected from Caco-2 cells incubated with APAP for 24 h, extracted and analysed alongside solutions of 5, 10 and 20 mM APAP to measure the percentage of APAP metabolism. Remaining concentrations of APAP were calculated from the percentage of original APAP remaining.

Targeted analysis of APAP and its metabolites in cell media samples collected from Caco-2 cells following 24 h culture with 5–20 mM APAP was carried out by LC-MS/MS as described in Section 2.4, alongside calibration standards aceta-minophen-d4 (APAP-d4; 10 ng) and APAP-SUL-d3 (10 ng).

## 4. Discussion

Following previous observations that even low (5 mM) doses of APAP disrupted hepatic tight junctions in HepaRG in vitro cell culture [14], we investigated the possible effects of APAP on intestinal epithelial cells and therefore indirectly on hepatocellular toxicity, using the enteric Caco-2 cell line as an in vitro model of the intestinal epithelial barrier. Investigations showed that Caco-2 cell cultures exposed directly to 5–20 mM APAP maintained cell adhesion, cell–cell tight junctions and membrane integrity for at least 24 h. This is in complete contrast to our previous results in which direct treatment of HepaRG cells with 5–20 mM APAP for 24 h resulted in a rapid loss of membrane integrity and cell viability and ultimately cell death [14]. It was also shown that Caco-2 cells can metabolise APAP, principally by the sulfation and glucuronidation pathways. Finally, when Caco-2-preconditioned cell culture medium previously containing 5–20 mM APAP was transferred to HepaRG cells for a further 24 h, no loss of cell viability or membrane integrity was seen at any APAP concentration tested. This suggests that Caco-2 cells can metabolise or otherwise alter APAP to derivatives that are no longer toxic to hepatic cells.

Not only was Caco-2 cell–cell viability and cell–cell tight junction integrity maintained in the presence of APAP, but we saw a concentration-dependent increase in impedance at 4 kHz, which became significant at higher (20 mM) APAP concentrations. This suggests a closing of tight junctions and an increase in rigidity of adhesion molecules within the membrane. This is consistent with the increased Caco-2 cell membrane integrity and reduction in paracellular transport in the presence of APAP, as reported by Schäfer et al. [16]. If replicated in vivo, this could reflect the strengthening of the intestinal epithelial barrier as part of an adaptive response to APAP, whereby the gut modulates entry of potentially toxic molecules.

A novel aspect of this study has been the ability to monitor cell adhesion, tight junctions and membrane integrity in a non-invasive manner using ECIS (electric cell-substrate impedance sensing) [23,24]. Previously, tight junction integrity would have been assessed by histochemistry for zonula occludens-1 (ZO-1, also known as tight junction protein-1), which is involved in signal transduction at cell–cell junctions and is thus a marker for the presence of tight junctions. The disadvantage of this approach is that ZO-1 can only be measured at a single terminal endpoint, whereas ECIS, using a low-voltage alternating current applied through electrodes at the bottom of the cell culture well to measure changes in cell membrane impedance, enables the non-destructive correlation of cell adhesion, tight junction and membrane integrity over the full period of cell culture, including any drug treatments. Crucially, cellular impedance data have been correlated with ZO-1 histopathology, confirming the use of ECIS to monitor tight junction integrity [14,23].

The current study also showed significant Caco-2 cell metabolism of APAP, principally by the sulfation and glucuronidation pathways. Although the first pass effect for orally administered drugs is a well-known phenomenon, there is little information on the importance of intestinal enterocytes in the metabolism of drugs administered by the oral route before reaching the liver [25]. In general, the gastrointestinal tract is regarded as only a minor component of first pass metabolism, with the liver considered the predominant site for drug detoxification [26]. However, the current study identifies the prospective contribution of enteric epithelial cells in reducing the quantity of orally administered APAP reaching the liver, by showing that more than 75–80% of APAP was metabolised in vitro by Caco-2 cells, thereby decreasing the hepatocellular toxicity of the conditioned medium. These observations, if replicated in vivo, could have important implications for the exposure of hepatic parenchyma to APAP administered via the intravenous versus oral route.

Although extrapolation of in vitro observations to in vivo APAP hepatotoxicity represents a significant challenge, most in vitro studies agree that concentrations of APAP above 5 mM are associated with hepatotoxicity [27]. Previously, we have shown a significant concentration-dependent loss of total ATP and tight junctions in HepaRG cells treated directly with 5–20 mM APAP for 24 h [14]. However, as demonstrated here, when incubated with Caco-2-preconditioned medium previously containing 5–20 mM APAP, HepaRG cells showed no significant loss of viability or compromise of cell membrane integrity, albeit there is a non-statistically significant trend towards an increase in membrane capacitance at higher APAP concentrations. LC-MS/MS analysis of Caco-2-APAP supernatants showed that 75–80% of APAP was metabolised; however, the abrogation of APAP toxicity to HepaRG cells cannot be ascribed entirely to the reduction in APAP concentration, as the supernatant transferred from Caco-2 cells exposed to 20 mM APAP contained ~4.62 mM APAP after exposure, comparable to the 5 mM APAP concentration previously shown to disrupt HepaRG tight junctions [14]. The explanation for this difference is not immediately apparent; however, previous observations suggested that preconditioning of C3A hepatic cells with fulminant hepatic failure (FHF) serum stimulated bioenergetic metabolism and thus improved cell viability when re-exposed to FHF serum [28]. Similarly, co-culture of hepatic (C3A cells) with vascular endothelial cells (HUVEC) ameliorated the cytotoxic effect of paracetamol on hepatic cells [29]. It is therefore possible that it is not simply metabolism and reduction in APAP that are important in our experiments, but also that a number of paracrine signals and protective cytokines may be secreted by Caco-2 cells into the preconditioned medium, exerting a protective effect on cell adhesion and, ultimately, on hepatic cell viability.

There are several possibilities for in vivo mitigation of hepatotoxicity by the gut. Firstly, while most metabolism of APAP occurs in the liver, enteric intestinal cells have key cytochrome P450 enzymes and glutathione in sufficient quantity to conjugate and metabolise APAP [8,9]. Secondly, the gut microbiome most likely plays an important role in bioavailability and absorption of APAP, thereby determining the degree of hepatotoxicity [30,31]. Microbiota and microbiome-encoded enzymes are now seen as probable intermediate targets which can alter pharmacokinetics and effect clinical response, while transit time may also play a significant part in the absorbance of APAP, and future studies should take this into consideration.

As our study was limited to preconditioned media with 24 h incubation, further experimentation with microfluidics or co-cultures could shed more light onto the biological pathways of APAP metabolism within Caco-2 cells in a more time-sensitive model where immediate effects between the gut and liver can be observed as drug and metabolites pass through their respective barriers. As per most similar studies, the current work was also limited by the absence of microbiome that emerges as an important factor in absorption and transport of APAP and metabolites within the gut–liver axis [30,31].

Despite the abovementioned caveats, our study highlights the possible protective effect of intestinal epithelial barrier with regard to APAP hepatotoxicity. Intravenous APAP was first licensed for use in 2004 and several studies have concluded that there is no difference in efficacy of paracetamol given intravenously or orally after surgery [32,33]. However, despite intravenous APAP being more expensive than oral APAP, it is being used excessively and can increase the risk of hepatotoxicity where variables such as weight, pre-existing renal impairment or glutathione deficiency are not taken into consideration [32,33,34].

In conclusion, metabolism of APAP within the gut–liver axis encompasses various factors such as site-specific metabolism, absorption, microbiome, bile acid accumulation and conjugation with GSH. Our study shows that APAP hepatocellular toxicity is mitigated in vitro due to the capacity of enteric epithelial cells to metabolise APAP and highlights the possibility of intravenous APAP causing hepatotoxicity at lower concentrations than when delivered via the oral route.

## Figures and Tables

**Figure 1 jcm-12-03995-f001:**
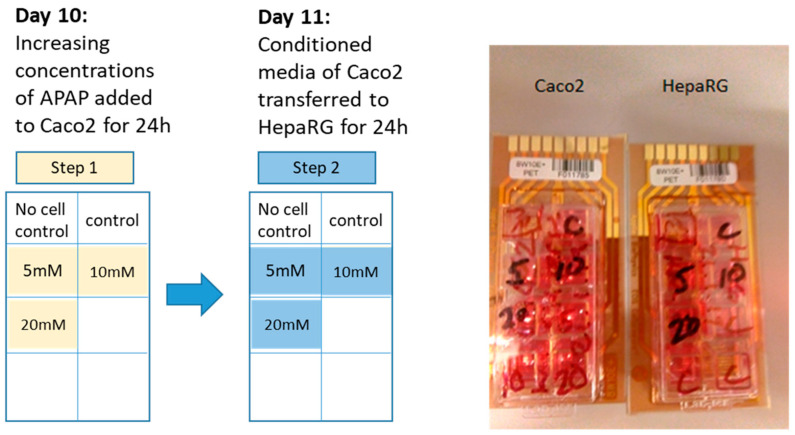
Stepwise media conditioning to mimic some aspects of enterohepatic circulation. Step 1: 5, 10 and 20 mM APAP was added to Caco-2 cells, alongside control, for 24 h. Step 2: Supernatant was collected from step 1 after 24 h and transferred to HepaRG cells for a further 24 h.

**Figure 2 jcm-12-03995-f002:**
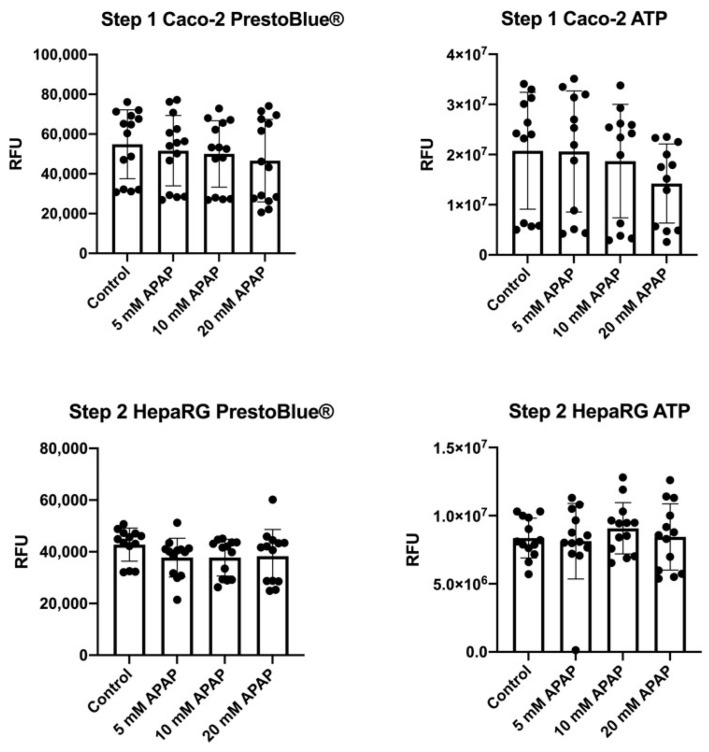
Cell viability. PrestoBlue^®^ and Promega CellTiter-Glo^®^ assays assessing mitochondrial health and ATP production in Caco-2 cells following direct treatment with 5–20 mM APAP for 24 h (Step 1) and HepaRG cells treated for 24 h with Caco-2-preconditioned medium that previously contained 5–20 mM APAP prior to 24 h culture with Caco-2 cells (step 2), *n* = 4 biological replicates with three technical replicates each.

**Figure 3 jcm-12-03995-f003:**
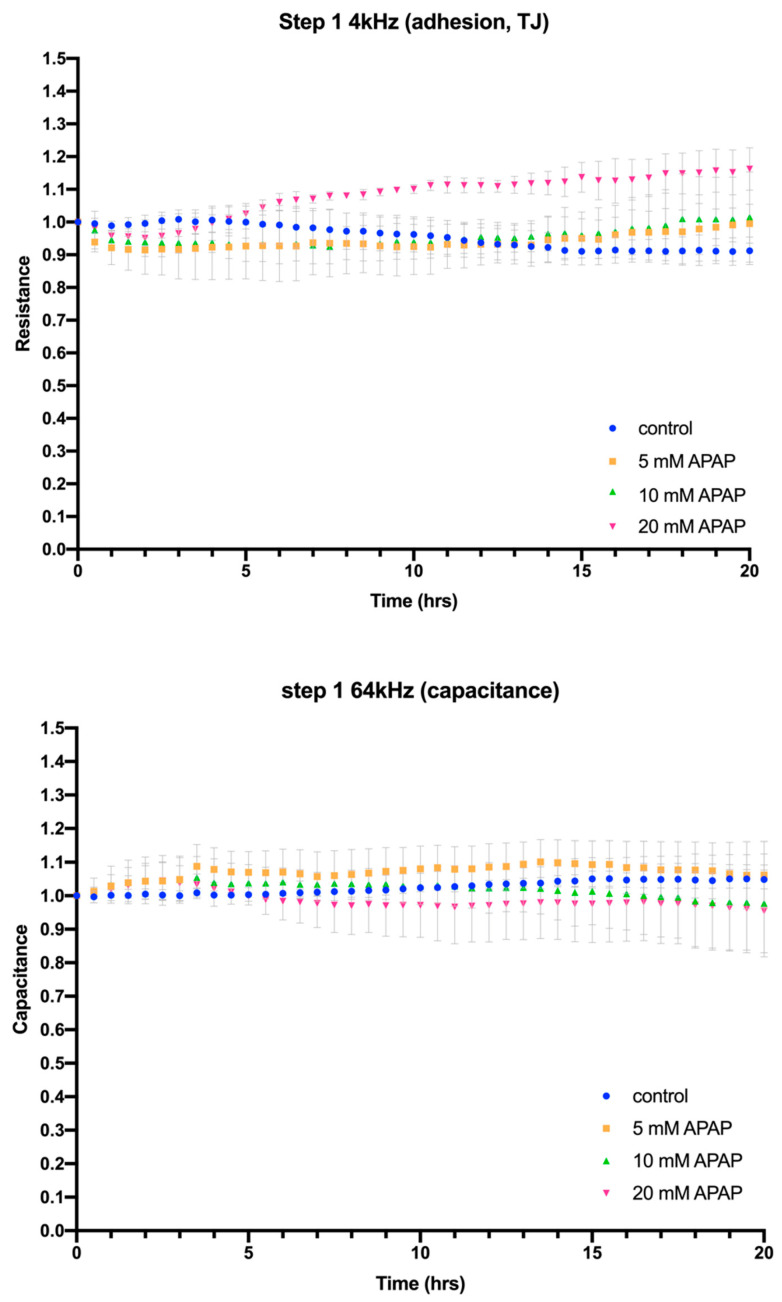
ECIS impedance-based assay of the effects on Cacao-2 cell adhesion, cell–cell tight junctions (4 kHz) and cell membrane integrity (64 kHz) of exposure to 5–20 mM APAP for 24 h (Step 1). Error bars in the resistance plots indicate the spread of starting resistance before normalisation of data points to enable comparison between individual biological replicates. Bar graphs show ECIS assay fold changes calculated from assay end points for cell adhesion and cell–cell tight junctions (4 kHz) and cell membrane integrity (64 kHz) compared to control, with a *p*-value of <0.05 indicated by an asterisk taken to be statistically significant; ns = not significant.

**Figure 4 jcm-12-03995-f004:**
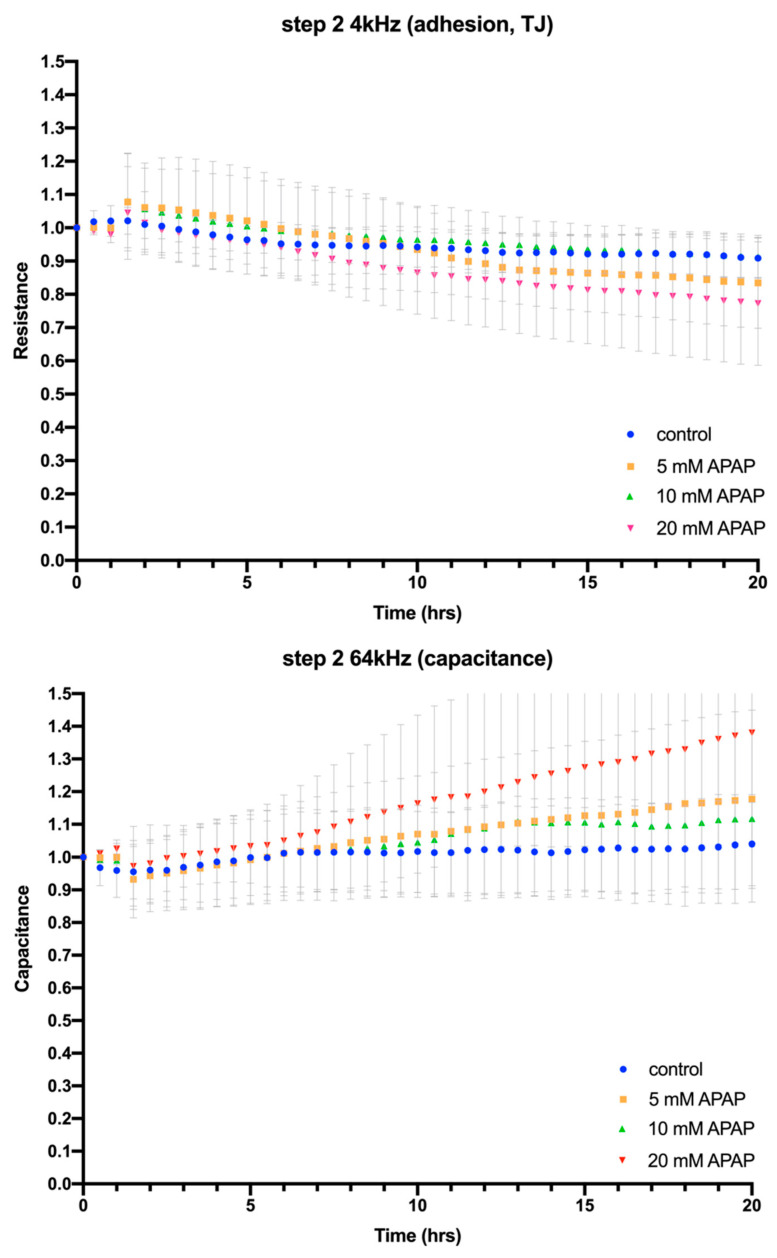
ECIS impedance-based assay of the effects on HepaRG cell adhesion, cell–cell tight junctions (4 kHz) and cell membrane integrity (64 kHz) of exposure to Caco-2-preconditioned medium for 24 h (Step 2). Error bars in the resistance plots indicate the spread of starting resistance before normalisation of data points to enable comparison between individual biological replicates. Bar graphs show ECIS assay fold changes calculated from assay end points for cell adhesion and cell–cell tight junctions (4 kHz) and cell membrane integrity (64 kHz) compared to control, with a *p*-value of <0.05 taken to be statistically significant; ns = not significant.

**Table 1 jcm-12-03995-t001:** APAP remaining after 24 h incubation on Caco-2 cells.

	Total APAP ng/Sample	Remaining APAP ng/Sample	Remaining APAP Percent/Sample	Remaining APAP mM
5 mM	116.11	19.36	16.67%	0.834 mM
10 mM	233.87	36.56	15.63%	1.56 mM
20 mM	378.00	87.72	23.21%	4.64 mM

**Table 2 jcm-12-03995-t002:** Metabolism of APAP via Caco-2 cells.

	APAP-Cys ng/mL	APAP-Sul ng/mL	APAP-Glu ng/mL
5 mM	129.9	1021	222
10 mM	220.95	1693	505.5
20 mM	480	926	732

## Data Availability

The data reported in this study have been deposited in the Edinburgh DataShare open access data repository at: https://doi.org/10.7488/ds/7471.

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
