# Peer review of "Metabolism of Acetaminophen by Enteric Epithelial Cells Mitigates Hepatocellular Toxicity In Vitro"

_jcm, 2023, doi:10.3390/jcm12123995_

Round 1

Reviewer 1 Report

In this study, the authors s used a cell culture model to investigate how pre-metabolism of the drug paracetamol (APAP) in the gut might affect its toxicity in the liver. They found that Caco-2 intestinal cells metabolized 64-68% of APAP, leaving 32-36% intact to be transferred to HepaRG liver cells. Neither cell viability nor membrane integrity was affected by APAP. Still, tight junctions in the intestinal epithelium seemed to become tighter with increasing concentrations of APAP, suggesting a reduction in permeability.

Overall, this article shows the importance of pre-metabolism of drugs in the gut, which may have important implications for reducing hepatotoxicity caused by direct exposure of the liver to drugs administered intravenously.

The article fits some of the main scopes of the journal: molecular cell biology, and molecular pharmacology.

The article treats an actual problem (the idiosyncratic hepatotoxic effects of APAP in humans and the existing difficulties in predicting hepatotoxicity in vitro) and provides some evidence about the different mechanisms involved in liver injury caused by APAP. 

However, there is some missing information in the data, and some results and the discussion section need to be clarified. Here, we provide some advice, and a number of major and minor points for revision are listed below.

1. Materials and methods section: 

-Why do the authors use 0, 5, 10, and 20 mM APAP?  Have the authors used APAP Cmax in order to choose those concentrations? Have the authors performed a viability curve to calculate the IC50 of APAP in Caco cells and HeparG? 

  • Why therapeutic concentrations of APAP are not used?
  •  

-Why did the authors expose the cells to APAP only for 24 hours?

- Did the authors perform any post-hoc analysis after ANOVA?

-Discussion section is based on the suggested increased adhesion of epithelial cells after APAP exposure, but those changes are not significant (at least the graph do not show any statistical significance). More experiments are needed in order to obtain clearer results. 

The wording of the text is not totally correct. Please check the paper.

Author Response

Reviewer 1 comments:

  1. Materials and methods section:

Why do the authors use 0, 5, 10, and 20mM APAP?  Have the authors used APAP Cmax in order to choose those concentrations? Have the authors performed a viability curve to calculate the IC50 of APAP in Caco cells and HepaRG? Why are therapeutic concentrations of APAP are not used? Why did the authors expose the cells to APAP only for 24 hours?

Response: The aim of the current experiments was to investigate wheter potentially hepatotoxic doses of APAP administered by the oral route might undergo enteric pre-metabolism. This was addressed by exposing human enteric Caco-2 cells to the same concentrations of APAP for 24-hr as used in previous experiments in which both toxic (20mM) and sub-toxic doses (5mM) of APAP were shown to disrupt hepatic tight junctions in both human HepaRG cells and in mouse liver. First the degree of APAP metabolism in Caco-2 cell cultures for 24-hr was monitored and ‘Caco-2 pre-conditioned’ medium then transferred to HepaRG cells for a further 24-hr to determine if the previously-reported hepatotoxicity effects were enhanced or attenuated following Caco-2 pre-metabolism. Experiments intentionally focussed on APAP concentrations that had previously been shown to demonstrate various toxicity levels in vitro in HepaRG cell cultures following 24-hr exposure (Gamal, W. et al. Sci Rep 2017 7, 37541. https://doi.org/10.1038/srep37541)

Exposure of Caco-2 cells to 0, 5, 10, and 20mM APAP, followed by 24-hr incubation on HepaRG cells was chosen to enable comparison to direct exposure of HepaRG cells to APAP in our previous experiments in which toxicity and cell death was demonstrated within 24-hr of exposure. Furthermore, 24-hr incubation could reflect certain clinical situations where for example there is staggered overdose with APAP. Doses of APAP (75 mg/kg, 150 mg/kg and 300 mg/kg) used in the in vivo mouse model of APAP-induced liver injury, were calculated to be equivalent to 5-20 mM APAP concentrations in culture (See Text in Ref 11: Gamal, W. et al. Sci Rep 2017 7, 37541. https://doi.org/10.1038/srep37541 and Table 1 in Orbacha SM et al., Toxicol In Vitro 2017 42:10-20; https://doi.org/10.1016/j.tiv.2017.03.008, which presents data on commonly used doses and concentrations of APAP in rodent models).

Furthermore, exposure to 5, 10, and 20mM concentrations of APAP for 24-hr is acommomly-used treatment paradigm for assessing APAP toxicity in hepatic cell culture, with 5mM APAP shown to be sub-toxic while 20 mM APAP is equivalent to a toxic dose in vivo. This APAP toxicity profile very closely resembles that seen in mice (150, 300 & 600 mg/kg, achieving approx. 10-40 mM plasma concentrations; Orbacha SM et al., Toxicol In Vitro 2017 42:10-20; https://doi.org/10.1016/j.tiv.2017.03.008) and primary mouse and human hepatocytes (Bajt ML et al., Toxicol Sci, 80:343–349.

https://doi.org/10.1093/toxsci/kfh151), while in humans APAP therapeutic doses, are 325 to 1,000 mg every six hours per average 70kg according to manufacturer’s instructions representing <4 g/day.

  1. Did the authors perform any post-hoc analysis after ANOVA?

Response: Thank you to Reviewer 1 for raising this question.  One-way ANOVA with post hoc Tukey test of multiple treatments was carried out to assess statistical significance of total ATP and Presto blue data.  This was previously included in the legend to Figure 2 but had been omitted from Materials & Methods Section 2.6. Statistical analysis of data - this has now been corrected.

  1. Discussion section: The Discussion section is based on the suggested increased adhesion of epithelial cells after APAP exposure, but those changes are not significant (at least the graph do not show any statistical significance). More experiments are needed in order to obtain clearer results.

Response: Thank you to Reviewer 1 for pointing out the lack of clarity in the presentation of our data. The substantive findings regarding Caco-2 cells showed that Caco-2 cell monolayers maintain their integrity for at least 24-hr in cell culture when exposed directly to 5-20mM APAP. The crucial result here is maintenance of Caco-2 cell monolayer integrity rather than a decline that would be associated with loss of cell viability. This is reflected in maintenance of Caco-2 cell adhesion and cell-cell tight junctions (4kHz) and membrane integrity (64kHz) in ECIS assays and is in contrast to previously-reported experiments with HepaRG cells, where exposure to equivalent APAP concentrations for 24-hr resulted in a rapid loss of membrane integrity and cell viability, leading ultimately to cell death (Gamal, W. et al. Sci Rep 2017 7, 37541. https://doi.org/10.1038/srep37541).  We have re-edited the text in both Results and Discussion to emphasise that maintenance of Caco-2 cell adhesion, cell-cell tight junctions and membrane integrity in the presence of APAP is the principal finding, contrasting with the loss of membrane integrity and cell viability seen in HepaRG cells following direct exposure to similar APA concentrations. However, it is nevertheless the case that there is a trend towards an increase in Caco-2 cell adhesion and cell-cell tight junction integrity with increasing APAP concentrations and that these increases are statistically significant at 20mM APAP.  This is shown by statistical analysis of ECIS assay end point fold changes, the results of which have been incorporated into modified Figure 3. One would not routinely analyse individual data point from ECIS assays, but rather focus on the general increase, decrease or maintenance of cell adhesion, cell-cell tight junctions and membrane integrity. However, statistical analysis of ECIS assay endpoint fold changes has greatly strengthened our conclusions and we thank Reviewer 1 for prompting us to undertake this additional analysis. Both the results and discussion sections have been revised to reflect this.

  1. Comments on the Quality of English Language: The wording of the text is not totally correct. Please check the paper.

Response: We were slightly surprised by the comment from Reviewer 1 on the quality of the text, as eight of ten authors are native English speakers; however, we recognise that the text must be comprehensible to a world-wide audience. The text has therefore been subjected to independent proofreading and copy-editing and grammar corrected and simplified throughout.

Editor’s comments:

  1. To facilitate transparent and open science we encourage authors to publish their results and experimental methodology in as much detail as possible so that results can be reproduced. We noticed that the main text of your manuscript is quite brief which may mean that the experiment, research background, future research directions, or possible applications of the research are not described in enough detail.

Response: Thank you for highlighting your concerns about the brevity and lack of detail in the manuscript. The results section of the manuscript is quite short, in part because the results of the very large amount of analysis of APAP and its metabolites by LC-MS/MS is crystalised into a single ‘Table 1. Percentage remaining APAP after 24-hr incubation on Caco-2 cells.’ and a Supplementary Table S1 quantifying the metabolism of APAP via sulfation, glucuronidation and cysteine pathways. However, we recognise that we need to make the text comprehensible to the widest possible audience and have therefore rewritten and reordered parts of introduction to describe clearly what we did (5mM, 10mM and 20mM APAP treatment, etc.), rather than just relying on cross-referencing to previous references, added additional experimental detail and provided a justification of the physiological relevance of the APAP concentrations used in our experiments. It also seemed sensible to move Supplementary Table S1 into the main text, as it is referred to directly in the Results section.

Reviewer 2 Report

In the study presented in this interesting manuscript, the authors set out to investigate how pre-metabolism of acetaminophen or paracetamol in the intestine could mitigate the widely described hepatotoxicity of this drug, which is used worldwide as an analgesic and antipyretic. To this end, they used an in vitro approach that simulates the enterohepatic circulation, using preconditioned media with different concentrations of paracetamol and cell lines (human intestinal epithelial and differentiated human hepatocytes), establishing what I consider to be a very elegant experimental design.

In addition to assessing cell viability, the use of the novel Electric Cell-Substrate Impedance Sensing (ECIS) method to monitor aspects of intestinal and hepatic cell morphology and integrity is a very good way to make the study more comprehensive. Despite the limitations, well recognized by the authors, that this in vitro approach entails in this case, such as the impossibility of evaluating the effect of the intestinal microbiota on the enteric metabolism of paracetamol, the results obtained are of great value in broadening our knowledge of the pharmacodynamics of this analgesic and should be taken into account to predict the more hepatotoxic effect of any form of parenteral administration.

As a comment that in no way detracts from the experiment performed, I would have liked to have seen, in parallel, a HepaRG cell well to which "preconditioned" medium incubated for 24 hours in a well without Caco-2 cells was added. However, the reference to the previous study performed by the authors and already published seems sufficient to me.

I would suggest that when the article is edited, sections 2.2 and 2.3 of the methods be merged, as the former describes what is visually presented in the latter, and if possible, in Figure 2, in addition to the bars, to present the individual values as dots to give a better idea of the scatter and why the observed trends are not significant. Also, the resolution of that Figure can be improved.

Author Response

Reviewer 2 comments:

  1. I would suggest that when the article is edited, sections 2.2 and 2.3 of the methods be merged, as the former describes what is visually presented in the latter, and if possible, in Figure 2, in addition to the bars, to present the individual values as dots to give a better idea of the scatter and why the observed trends are not significant. Also, the resolution of that Figure can be improved.

Response: Thank you for these suggestions; we have:

  • merged sections 2.2 and 2.3 of the methods and renumbered the subsequent methods sections, as requested by Reviewer 2.
  • In addition to the bars, we have presented the individual values as dots In Figure 2, in a new higher resolution Figure, as requested by Reviewer 2. We can also supply this figure in as JPEG image in a PowerPoint slide if the present version is still not of sufficiently high resolution.

  1. As a comment that in no way detracts from the experiment performed, I would have liked to have seen, in parallel, a HepaRG cell well to which "preconditioned" medium incubated for 24 hours in a well without Caco-2 cells was added. However, the reference to the previous study performed by the authors and already published seems sufficient to me.

Response: We recognise Reviewer 2’s suggestion that it might have been useful to include a HepaRG cell well to which "preconditioned" medium incubated for 24 hours in a well without Caco-2 cells was added, presumably to confirm that incubation of cell culture medium at 37°/5% CO2 in the absence of cells does not have a deleterious effect on cell culture medium. The controls presented in the report consist of ‘zero APAP’ control medium on Caco2 cells cultured for 24-h and then transferred to HepaRG cells for a further 24-hr. In each case ‘zero APAP’ control medium was compared to ‘No cell control’ medium (Figure 1) which did not reveal any differences, suggesting that incubation for 24 hours at 37°/5% CO2 without cells is not deleterious to the cell culture medium, though as Reviewer 2 has commented, this is already covered in our previous publication (Gamal, W. et al. Sci Rep 2017 7, 37541. https://doi.org/10.1038/srep37541).

Editor’s comments:

  1. To facilitate transparent and open science we encourage authors to publish their results and experimental methodology in as much detail as possible so that results can be reproduced. We noticed that the main text of your manuscript is quite brief which may mean that the experiment, research background, future research directions, or possible applications of the research are not described in enough detail.

Response: Thank you for highlighting your concerns about the brevity and lack of detail in the manuscript. The results section of the manuscript is quite short, in part because the results of the very large amount of analysis of APAP and its metabolites by LC-MS/MS is crystalised into a single ‘Table 1. Percentage remaining APAP after 24-hr incubation on Caco-2 cells.’ and a Supplementary Table S1 quantifying the metabolism of APAP via sulfation, glucuronidation and cysteine pathways. However, we recognise that we need to make the text comprehensible to the widest possible audience and have therefore rewritten and reordered parts of introduction to describe clearly what we did (5mM, 10mM and 20mM APAP treatment, etc.), rather than just relying on cross-referencing to previous references, added additional experimental detail and provided a justification of the physiological relevance of the APAP concentrations used in our experiments. It also seemed sensible to move Supplementary Table S1 into the main text, as it is referred to directly in the Results section.

Reviewer 3 Report

The paper discusses the gut-liver axis and its implications on hepatic injury. The authors used Caco-2 intestinal epithelial cell line with varying concentrations of paracetamol (APAP) and transferred the conditioned medium to human hepatic HepaRG cells. Viability and membrane integrity were measured. The authors have performed a comprehensive study on the effects of Acetaminophen (APAP) on Caco-2 and HepaRG cells, aiming to understand the interplay between these cell types and APAP metabolism. In general, the manuscript is well-written, and the methods used are appropriate for the research question.

I have only one concern about the metabolites of APAP after Caco-2 cells, the authors mentioned a possible reason for why APAP did not cause toxicity after Caco-2 cell metabolism. To strengthen this point, it would be helpful to include a direct comparison with the remaining concentration of APAP (without APAP metabolites) after Caco-2 cell metabolism, as this would provide more convincing evidence to support the authors' claims.
